# Central Giant Cell Granuloma in the Mandibular Condyle in a Teenager. A Case Report with Literature Review

**DOI:** 10.3390/jcm11144239

**Published:** 2022-07-21

**Authors:** André Luís Costa Cantanhede, Sergio Olate, Adriano Freitas de Assis, Márcio de Moraes

**Affiliations:** 1Division of Oral and Maxillofacial Surgery, Piracicaba Dental School, State University of Campinas, Piracicaba 13414-903, SP, Brazil; andre_ctbmf@hotmail.com (A.L.C.C.); mmoraes@fop.unicamp.br (M.d.M.); 2Division of Oral and Maxillofacial Surgery & CEMYQ, Universidad de La Frontera, Temuco 4780000, Chile; 3Department of Oral and Maxillofacial Surgery, Bahiana School of Medicine and Public Health, Salvador 40290-000, BA, Brazil; adrianoassis@hotmail.com

**Keywords:** giant cell granuloma, mandibular condyle, orthognathic surgery, TMJ

## Abstract

Central giant cell granulomas (CGCG) are not common in the mandibular condyle. In teenagers, the problem is more complex because of difficulties in diagnosis and treatment involving the potential growth of the mandibular process and development of the face. In this short communication a case is presented of an eleven-year-old female under diagnosis of central giant cell granuloma affecting the mandibular condyle treated surgically in two steps using a condylectomy and vertical ramus osteotomy at the first time and later orthognathic surgery, showing the clinical evolution after 13 years of follow-up. In addition, we performed a review of the scientific reports related to CGCG in the mandibular condyle to compare this treatment with others, in terms of follow-up and results. We concluded that the CGCG affecting the mandibular head can be properly treated with low condilectomy, vertical mandibular ramus sliding osteotomy, and discopexy.

## 1. Introduction

Central giant cell granuloma (CGCG) is an intraosseous lesion with three main features: the presence of fibrous connective tissue with multinucleated giant cells, hemorrhagic foci, and bone trabeculation [1,2,3,4]. It comprises close to 7% of all intraosseous lesions in the jaw; 60% of cases affecting patients under 30 years old, frequently in the anterior area of the mandible [5,6,7,8].

Jaffe in 1957 first described this lesion in the maxillofacial region as a reparative intraosseous lesion secondary to trauma, inflammatory conditions, or hemorrhage [7]. Its location in the mandibular head show a low frequency, thus making the diagnosis and therapy a challenge for the surgeon due to the restricted access and functional impairment that can arise from the treatment.

The aim of this short communication is to present a case of CGCG involving the mandibular condyle in a young patient, 11 years old, treated by surgery in a two-stage strategy with 13-year follow-up and to perform a review of the treatment and follow-up of published cases.

## 2. The Subject

A 11-year-old Caucasian female patient was referred to the Department of Oral and Maxillofacial Surgery by the orthodontist in 2008. In the initial exam was observed a well-defined radiolucency in the orthopantomography in the right mandibular condyle. Clinically, the patient showed a normal open mouth with no restriction in mandibular movement, without any increase in facial volume; however, she complained of mild pain in chewing. The symptoms started one year earlier, after a trauma on the chin with no fractures. Computer tomography (CT-scan) showed a well-defined hypodense area involving the right mandibular condyle, without cortical perforation (Figure 1). In the same line, the patient showed an orthodontic treatment with no dental extraction at the initial diagnosis with a slight mandibular retrognathia.

The initial diagnosis ranged from the simple bone cyst, aneurysmal bone cyst, CGCG, or intraosseous hemangioma. Laboratory tests for calcium, phosphate, alkaline phosphatase and parathyroid hormones were within a normal range.

The treatment plan was assessed by a 3D model using a classic preauricular and retromandibular approaches under general anesthesia with nasotracheal intubation. Curettage of the lesion was performed, and an intraoperative frozen section was conducted, ruling out the presence of malignant cells. Thereafter, a low condylectomy was carefully performed; after that, a vertical ramus osteotomy from the sigmoid notch to inferior mandibular border was realized, with stability in the alveolar inferior bundle.

The position of the mandibular ramus obtained from the vertical osteotomy was sliding and fixed in an upper level using a surgical guide, creating a “neo-condyle” to replace the condylar head. Osteotomy was fixed with two 2.0 mm L-shaped titanium plates and 8 mm long screws (Figure 2). A new contouring was performed in the lower border and the discopexy was realized using the native articular disc over the remaining sliding stump with 2.0 Ethibond sutures. The patient was discharged from hospital the following day. Posterior histopathological analysis of the entire specimen revealed the CGCG. Not included in this approach was any dental change, because of the urgent requirement for the biopsy, the early age of this patient and the potential growth after the first surgical approach. The patient was followed up closely after the first surgical time, and 3 years later the mouth opening was 39 mm, and no complication was observed (Figure 2).

One year after the first surgery, a new analysis was performed to confirm the dentofacial condition and on this occasion the orthognathic surgery was included. Dental occlusion was treated using orthodontics with dental extraction to create a class II dental occlusion (Figure 3). After 4 years, at 16 years old, the patient was submitted to a second operation with bimaxillary orthognathic surgery.

The orthognathic surgery was realized to modify the maxillomandibular position and the skeletal class II malocclusion, using a bilateral mandibular sagittal split osteotomy for advancement movement and the Le Fort I osteotomy for upper movement. The chin was not included at this moment. After 13 years from the initial treatment, the patient remained asymptomatic, without functional disturbances, and disease-free (Figure 4); the chin could be involved in a third step to obtain better symmetry and an aesthetic mandibular contouring due to facial asymmetry, but the patient refused this treatment at the moment; it should be mentioned that the orthognathic surgery and TMJ condition could be included in a revision time if necessary. However, to the best of our knowledge, this is the case with CGCG in this location with the longest follow-up.

## 3. Discussion

The World Health Organization describes CGCG as a bone lesion [4,9,10]. The treatment is controversial (Table 1), and to the best of our knowledge this is the first well-documented case with the longest clinical follow-up treatment of a CGCG in the mandibular condyle described up to now.

The first case published of peripheral CGCG in the head of the mandible was in a 44-year-old woman [1]. Choung et al. [9] divided these tumors into aggressive and non-aggressive variants; the aggressive variant is commonly seen in the young population, with a progressive localized volume increase and pain involved in 20% of cases [3,5,9].

CGCG histologically it looks like an aneurysmatic bone cyst and the absence of blood-filled gaps in the CGCG is an important feature for diagnosis [13]. In the case of cherubism and brown tumor secondary to hyperparathyroidism, the clinical and laboratory findings are important for diagnosis [4].

Treatment strategies for the CGCG range from systemic to local treatment; systemic included the use of calcitonin, hypodermic interferon-alpha 2a, intralesional steroid injections [13] and subcutaneous denosumab [8]. However, surgical intervention remains the most widely used therapeutic modality, mainly in aggressive lesions of the jaws, which might range from surgical curettage to bone resection [8,11,13].

The medical treatment included for this lesion mainly involves pharmacology applied directly into the lesion; intralesional corticosteroid has been used as well, showing variability in the long-term follow up [14,15,16]. Denosumab and immunotherapy have been used as neoadjuvants for these cases [17]. Denosumab has been included with good results [17,18]; however, in the long term follow up there is some controversy over the results and stability of the treatment [18,19]. Unfortunately, the main articles published for these conditions are case reports or case series with short- or medium-term follow up.

In the opinion of the authors, the side effects of pharmacological treatment, the location in the mandibular condyle, the involvement of facial condition and dental occlusion and the precision required for diagnosis suggest surgical treatment as the best course of action.

Strategies for condylar reconstruction described in these situations include autogenous grafts (e.g., costochondral) or alloplastic materials [3,7]. Teenagers show great potential for facial growth between 10 to 15 years old [20], and for that reason autogenous bone could be the best choice in these patients. The evolution of the adaptative new condyle could be important in the facial development. TMJ prosthesis has been used in cases of young people, showing good results as well [21], but this case was treated in 2008, within the initial evolution for TMJ replacement in young people.

Reconstructions including autogenous donors show differences in long-term results. Parmar et al. [22] reported the vertical ramus osteotomy and temporalis myofascial flap in 10 patients showing stability and close to normal mouth opening after 2 years follow up in subjects with TMJ ankylosis. On the other hand, Al-Moraissi et al. [23] performed an extensive review of TMJ reconstructions, showing that the use of costochondral grafts could be related to recurrence in TMJ ankylosis and the results in maximal inter-incisal opening would be lower when compared to other techniques.

Nowadays, TMJ reconstruction with alloplastic materials is safe for TMJ replacement, with patient-specific implant (PSI) strategies being recommended in some reports [24]. Meurechy and Mommaerts published a review with a focus on the historical evolution of this treatment, materials to build the system and indications for treatment, concluding that customized TMJ replacement is effective. Previously, Meruri [25] showed the indications for TMJ replacement including the reconstruction of the condylar unit. Bach et al. [26] showed the use of TMJ replacement in 348 patients, using PSI in cases of arthritis, ankylosis and degenerative TMJ pathology, presenting good results in the follow up.

The age of the patient and the requirement for secondary or tertiary surgical treatment in this patient were the main reason for the use of an autogenous reconstructive surgery using the mandibular ramus as the donor site. In 2013, Sidebottom [27] reported that the potential for growth in the child is a reason to choose autogenous reconstruction, leaving the possibility of changing to PSI TMJ replacement in the future.

Orthognathic movement in an 11-year-old could show unexpected evolution [20]; for that reason, primary reconstruction of the TMJ would be the best choice in the first instance, showing biological adaptation and function with disc reposition, muscle work and stability in mandibular movement. The secondary treatment for this patient was related to dental preparation, using dental extraction to permit mandibular movement and orthognathic surgery and orthodontics to treat dental occlusion, function and facial morphology.

## 4. Conclusions

Though this is an isolated case which may not be applicable to patients in general, we showed that CGCG affecting the mandibular head can be properly treated with low condilectomy, vertical mandibular ramus sliding osteotomy, and discopexy. In addition, orthognathic surgery can be carried out later to treat remaining dentofacial deformities.

## Figures and Tables

**Figure 1 jcm-11-04239-f001:**
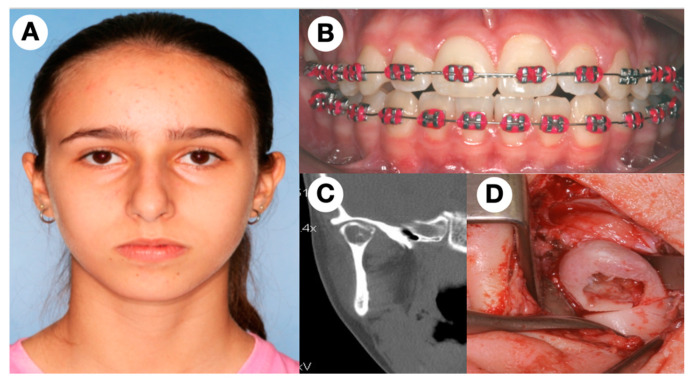
(**A**) Frontal view of an 11-year-old female patient with no asymmetry. (**B**) Dental occlusion at the moment of CGCG diagnosis using a conservative technique with no requirement for orthognathic surgery at that moment, (**C**) CT of the TMJ showing the area involved in the head of the condylar process, (**D**) Surgical treatment showing the CGCG with the lower osteotomy to remove the condylar head.

**Figure 2 jcm-11-04239-f002:**
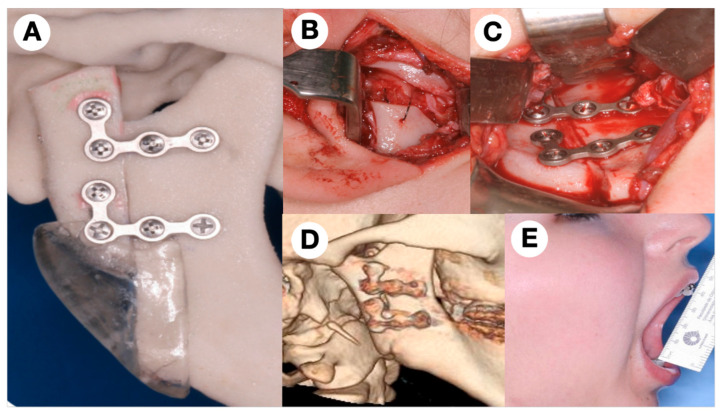
(**A**) 3D model used to perform the vertical osteotomy and the surgical guide (technology used in 2008), (**B**) discopexy used over the “new condyle” after upper reposition of the sliding osteotomy, (**C**) fixation of the vertical osteotomy using 2.0 type L plate, (**D**) CT of the area after 1 year from the first surgery showing a good repair and bone stability, (**E**) mouth opening after 1 year was close to 40 mm with no pain.

**Figure 3 jcm-11-04239-f003:**
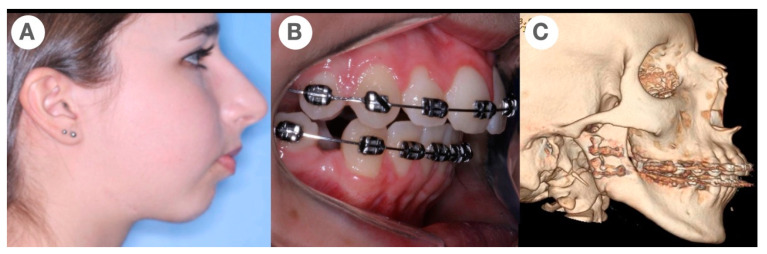
(**A**) Facial profile 4 years after the first surgical approach (16 years old) (**B**) dental occlusion of Angle class II, using bicuspid extraction to confirm facial and dental class II, (**C**) CT used to confirm mandibular position and to perform the skeletal analysis.

**Figure 4 jcm-11-04239-f004:**
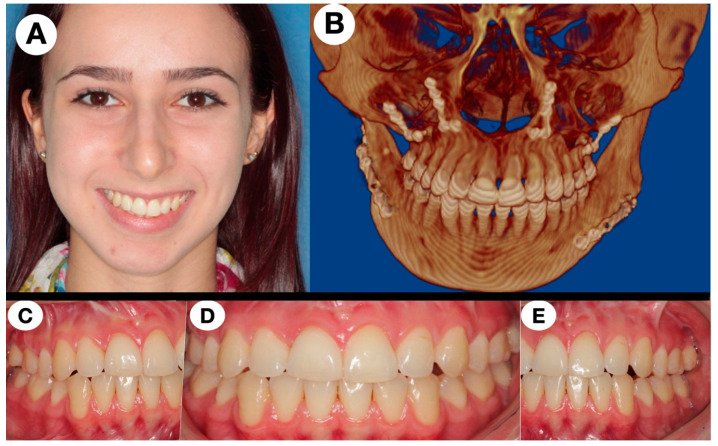
(**A**) Frontal view after 13 years from the first surgery showing asymmetry in the chin, good movement for muscle and regular lines for smile, (**B**) CT to confirm the regular position of the fixation used for orthognathic surgery, (**C**) right view of dental occlusion at the final control, (**D**) frontal view of dental occlusion at the final control, (**E**) left view of dental occlusion at the final control. The patient was free of pain and morbidity.

**Table 1 jcm-11-04239-t001:** Summary of the cases of Central Cell Giant Granuloma in mandibular condylar head reported in English literature.

Author/Year	Age/Sex	Clinical Findings	Imaging	Treatment	Follow-Up
Tasanen et al. (1978) [10]	59/M	Right painless slow-growing pre-auricular swelling. Mouth-opening limitation	Well-defined multilocular radiolucency—25 mm (right condyle)	Resection + reconstruction with CCG	21 months NRD
Shensa/Nasseri, (1978) [11]	15/M	Expansive mass. Asymptomatic	Well-defined radiolucency	Enucleation	NI
Abu-El-Naaj et al. (2002) [5]	15/F	Left preauricular swelling (2 months). Asymptomatic. Normal mandibular motion	Well-defined unilocular radiolucency (3 × 2 cm), left condyle	Enucleation	6 months NRD
Özcan et al. (2005) [1]	44/F	Right preauricular swelling (2-year evolution). Pain	CT—hypodense 2 × 2 cm. MRI rim-like hypodense T1. Peripheral calcification	Curettage	1 year NRD
Sun et al. (2009) [12]	NI (1 out of 22 cases)	Pain and tenderness	Well-defined radiolucency. Extending to the coronoid process	Curettage	NI
Jadu et al. (2011) [2]	31/M	Painful slow-growing left pre-auricular swelling (2 years). Limited mouth opening	CT—Well defined, expansile (3.5 cm), with a granular bone pattern	Enucleation then resection after recurrence	4 years NRD
Munzenmayer et al. (2013) [6]	19/F	Asymptomatic. No occlusal or mandibular movement disorders	Well-defined multilocular radiolucency (4 × 2.2 × 1.5 cm), with granular bone pattern (left condyle)	Resection + reconstruction with NVFF	2 years NRD
Gigliotti et al. (2015) [13]	29/M	Firm and nodular left pre-auricular swelling (6 months). No occlusal changes. Discomfort during the mandibular function.	Left multilocular radiolucency with thin cortices 5.5 × 3.8 × 3.4 cm	Resection + reconstruction with CCG	1 year NRD
Chang et al. (2016) [7]	37/F	Hardened left pre-auricular mass 4 × 3 cm. Asymptomatic	Well-defined radiolucent and radiopaque areas involving two cortical in left condyle + ramus	Resection + reconstruction with CCG	10 years NRD
Pai et al. (2017) [8]	2/M	Progressive and firm increase in the right preauricular region. Painless. (4 months)	Expansive multinucleated radiolucent lesion (4.2 × 3.5 × 4 cm) with cortical perforation.	Resection + reconstruction with CCG	18 months NRD
Khanna et al. (2018) [3]	22/M	Right preauricular swelling (4 months). Pain and TMJ movements restricted	Large lesion extending to the coronoid with areas of cortical perforation at multiple places.	Resection + 2.4 mm reconstruction plate with condylar head	1 year NRD
Bocchialini et al. (2020) [4]	60/F	Right preauricular pain (1 year). Mouth-opening limitation.	A large radiolucent lesion with distortion of the right condyle	Enucleation	1 year NRD
*Present case*	11/F	No swellings, joint pain during mandibular motion (1 year)	Well-defined radiolucent lesion on the right mandible head	Resection + sliding vertical ramus osteotomy + later orthognathic surgery	13 years NRD

Legend: M (male), F (female), CT (computer-tomography), MRI (Magnetic resonance imaging), CCG (costochondral graft), NVFF (non-vascularized fíbula flap), NRD (no-recurrence described), NI (not informed).

## Data Availability

The study did not report any data.

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
