# Peer review of "Central Giant Cell Granuloma in the Mandibular Condyle in a Teenager. A Case Report with Literature Review"

_jcm, 2022, doi:10.3390/jcm11144239_

Round 1

Reviewer 1 Report

The topic is a interesant one. Some things are to be modified. 

Firstly, the authors insist about malocclusion of the patients, but are missing the preoperative x-rays (especially cephalograms!) and the photos . Some comments are necessary about the occlusal plan, why they decided not to be corrected. In this case, is a direct connection with the condylar problem.

On the other hand, they talk about a review. Discussions chapter must be significant augmented. A special focus must be addressed to different possibilities of TMJ reconstructions.  And last but not least, they talk something about medical method to treat the giant cell granuloma. But don't tell nothing about these method. 

I wait to hear soon, because topic worth the effort for a good article

Reviewer 2 Report

The clinical case study addresses an interesting topic for the scientific sector and allows for the verification of the long-term efficacy of a demolitive surgical treatment of the condyle in the age of growth and reconstruction with autologous bone. The clinical case is well documented except for the need for some minor revisions.

I think it is useful to insert the photo of the dental occlusion in the post-operative phase of the first operative step and pre-and post-surgical cephalometric analysis of the second operative step. As for the second surgical stage, it is unclear how many years later it was performed and could be better specified in line 82.

Line 16” the a scientific reports” 

Line 48 “was diagnosis” to be corrected

Figure 1: caption to be corrected

 a) "dental occlusion” this image is not present in the document

 b) "CT of TMJ” this is not a TC image

c) "surgical treatment” this photo is not present in the document

      Line 55 “The initial diagnostic...” to be corrected

Line 92 “(table1)” there is no table in the text

Reviewer 3 Report

This is an interesting case report of a central giant cell granuloma in the mandibular condyle, treated surgically and supplemented by orthognathic surgery. The article needs revision. Moderate English changes are required.

Do not use the term "this pathology" or "bone pathology"; it is a bone lesion or bone tumor.

At the end of The subject, it is mentioned that "the chin would be involved in a third surgery..." but it should be mentioned that the orthognathic surgery should also be submitted to revision, as there was vertical asymmetry of the jaws.

Please review the captions for figures. In Figure 1 letters do not correspond to the images shown. Figure 2, letter e, does not refer to open bite but to mouth opening.

In the Discussion cite the prognosis of this bone lesion. And what are the advantages of the graft used from the mandibular ramus relative to the costochondral graft?

Reviewer 4 Report

Dear Authors

the topic of the manuscript is interesting. The text is well written.

I only have some suggestions to improve it.

- It should be advisable to add intraoral figures at the biginnig of treatment;

- Please better describe the orthodontic preparation for surgery (line 81);

- line 81 correct orthodontic with Orthodontic (capital letter);

- it woud be advisable to add intraoral pictures (right and left view) at the end of treatment to better understand the occlusal results;

- Authors state that a third surgery to obtain better simmetry was advisable (line 82-83). In my opinion figure 3 frontal view shows that it is not only the chin to present asymmetry but also the mandible that is rotated to the right side. Shoud have Authors taken into consideration to overcorrect this asymmetry by an overcorrection of the mandibular sagittal split osteotomy to the left side? 

Please address these concerns.

Best regards

Round 2

Reviewer 4 Report

Dear Authors

all the required modifications have been addressed. The manuscript now has been improved.

Best regards